# A Review of Dynamic Object Filtering in SLAM Based on 3D LiDAR

**DOI:** 10.3390/s24020645

**Published:** 2024-01-19

**Authors:** Hongrui Peng, Ziyu Zhao, Liguan Wang

**Affiliations:** 1School of Resources and Safety Engineering, Central South University, Changsha 410083, China; 215512098@csu.edu.cn (H.P.); csu_zzy93@csu.edu.cn (Z.Z.); 2Changsha Digital Mine Co., Ltd., Changsha 410221, China

**Keywords:** SLAM, LiDAR, dynamic point cloud filtering

## Abstract

SLAM (Simultaneous Localization and Mapping) based on 3D LiDAR (Laser Detection and Ranging) is an expanding field of research with numerous applications in the areas of autonomous driving, mobile robotics, and UAVs (Unmanned Aerial Vehicles). However, in most real-world scenarios, dynamic objects can negatively impact the accuracy and robustness of SLAM. In recent years, the challenge of achieving optimal SLAM performance in dynamic environments has led to the emergence of various research efforts, but there has been relatively little relevant review. This work delves into the development process and current state of SLAM based on 3D LiDAR in dynamic environments. After analyzing the necessity and importance of filtering dynamic objects in SLAM, this paper is developed from two dimensions. At the solution-oriented level, mainstream methods of filtering dynamic targets in 3D point cloud are introduced in detail, such as the ray-tracing-based approach, the visibility-based approach, the segmentation-based approach, and others. Then, at the problem-oriented level, this paper classifies dynamic objects and summarizes the corresponding processing strategies for different categories in the SLAM framework, such as online real-time filtering, post-processing after the mapping, and Long-term SLAM. Finally, the development trends and research directions of dynamic object filtering in SLAM based on 3D LiDAR are discussed and predicted.

## 1. Introduction

SLAM (Simultaneous Localization and Mapping) is a cutting-edge technology that depends on specific environmental sensors to achieve self-localization in a given environment while simultaneously creating a model of the surrounding scene. This innovative technology has gained widespread recognition and adoption across several domains, including autonomous robots [1] and various intelligent fields. A typical SLAM system consists of a front-end module and a back-end module [2,3]. The front-end module collects data such as LiDAR (Laser Detection and Ranging) point clouds and camera images from sensors and finds the transformation between consecutive data frames, and the back-end module corrects the front-end estimation drifts by performing the loop closures [4].

SLAM based on 3D LiDAR is a technique that uses 3D laser scanners to perform simultaneous localization and mapping of an environment. 3D LiDAR is an active sensor that emits laser beams in multiple directions and measures the travel time and the phase difference of the reflected beams to create a 3D point cloud of the environment. The point cloud represents the spatial layout of the environment, which can be used to build a map and localize the robot within it. As research in the field of 3D LiDAR continues to advance [5,6], the exploration of SLAM technology based on this has gained significant momentum across various domains [7,8,9,10,11,12,13,14,15,16,17]. Consequently, numerous remarkable algorithms have been developed [18,19,20,21,22,23,24,25,26], with the most widely adopted ones being LOAM(LiDAR Odometry and Mapping) [20], and its various variants [2,21,22]; LIO (LiDAR-Inertial Odometry) [23,24,27,28]; and R3LIVE [25,26].

Notably, mainstream SLAM algorithms and point cloud alignment methods [29,30] are primarily developed under the premise of a static environment, while inadequately taking into account dynamic objects in the spatial realm. However, in real-world applications, an abundance of dynamic objects exists, such as pedestrians, vehicles, cyclists, temporary facilities or structures, and the like. In the street view shown in Figure 1, several common dynamic objects have been framed.

The presence of dynamic objects can make it inevitable that the scan data from 3D LiDAR includes representations of them [31,32,33,34,35], thereby significantly impeding the effectiveness of SLAM algorithm implementation, with three main detrimental effects as shown below:At the front-end level, the accuracy of point cloud alignment can be affected by the presence of dynamic objects, and even the robustness of SLAM alignment algorithms may be severely compromised if there is an excessive number of dynamic point clouds.At the mapping level, various approaches have been proposed to model 3D environments [36,37] or sub-sampled representations [38,39]. However, full resolution 3D point clouds form a basis for extracting useful information from the map [32]. The presence of a significant number of dynamic point clouds in the 3D point cloud can result in a plethora of *ghost-trail effects* [32,40], which may obscure or overlap with the static features in the scene. This can severely hamper the effectiveness of the mapping.At the map-based localization level, the presence of numerous dynamic point clouds in the priori map can result in significant deviations between the priori map and current observations, and even act as obstacles that can interfere with the localization performance [41,42,43]. These effects can also hurt long-term map maintenance.

As previously discussed, the presence of dynamic objects can substantially compromise the accuracy and robustness of SLAM in various real-world scenarios, such as navigating dense crowds [44] and urban autonomous driving [45], resulting in degraded performance and potential safety hazards. It can be seen that detecting and filtering dynamic objects is a critical problem to be addressed in SLAM. This work aims to summarize how SLAM based on 3D LiDAR can effectively filter out dynamic objects in changing environments. The entire pipeline is demonstrated in Figure 2 and the main contributions are summarized as follows:Regarding the research of filtering dynamic objects in SLAM based on 3D LiDAR, this work proposes a comprehensive examination from two dimensions: the solution-oriented level and the problem-oriented level. The former concentrates on exploring the diverse underlying principles behind specific methods for detecting dynamic targets in 3D point cloud. The latter, on the other hand, focuses on the application of dynamic point cloud filtering techniques in downstream tasks, investigating the efficient removal of various dynamic objects in real-world scenarios within the overarching framework of SLAM. It explores the adoption of processing strategies tailored to match the dynamic degree of these dynamic objects, enabling rapid and effective filtering. Despite their distinct natures, these two dimensions are interconnected, as the principles governing the methods used in studies with different processing strategies may coincide. Similarly, methods based on different principles can find application within the same processing strategy.According to the different principles of detecting dynamic targets in a 3D point cloud, the dynamic point cloud filtering methods fall under different classifications, such as the ray-tracing-based approach, the visibility-based approach, the segmentation-based approach and others, and the main idea and problems of various methods are summarized.The classification of dynamic objects in the real-world according to their degree of dynamics and their different processing strategies, from online real-time filtering to post-processing after the mapping to Long-term SLAM for different categories of dynamic objects in the SLAM framework based on 3D LiDAR are presented, to help readers effectively choose a more appropriate solution for their specific applications.

**Figure 2 sensors-24-00645-f002:**
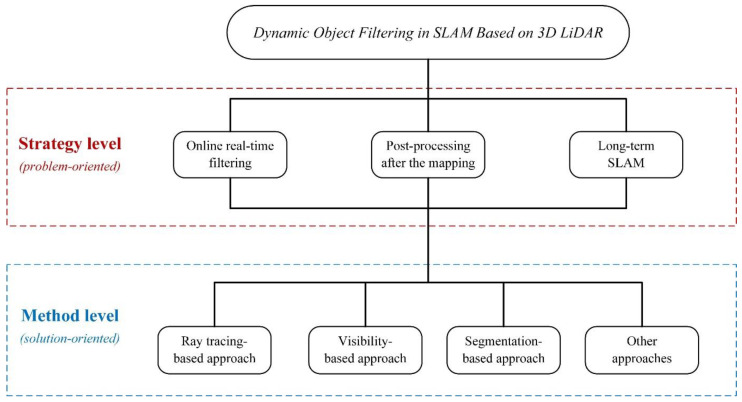
The pipeline of this paper. We present a summary of filtering dynamic objects in SLAM based on 3D LiDAR from two dimensions: the Method level and the Strategy level. The Method level is solution-oriented, we categorize the specific methods for filtering dynamic targets in a 3D point cloud into the ray-tracing-based approach, the visibility-based approach, the segmentation-based approach, and others, each based on distinct detection principles. The Strategy level is problem-oriented, and in conjunction with the SLAM framework based on 3D LiDAR, we propose three diverse processing strategies tailored to objects exhibiting different degrees of dynamics: online real-time filtering, post-processing after the mapping, and Long-term SLAM.

This paper presents a comprehensive overview of dynamic object filtering techniques in SLAM based on 3D LiDAR, divided into four parts. The first section emphasizes the necessity and importance of dynamic object filtering in SLAM. Subsequently, the paper is structured around the integration of dual dimensions. First, at the solution-oriented level, Section 2 elucidates a selection of mainstream methods based on their principles, along with the various research carried out using each method. Then, at the problem-oriented level, in Section 3 dynamic objects are classified according to their degree of dynamics, the processing strategies for different categories of dynamic objects in the SLAM framework are summarized, and the advantages and disadvantages of representative algorithms from different methods are analyzed and compared. Finally, Section 4 offers a summary of the entire paper and an outlook on future research directions. Table 1 provides a list of abbreviations used in this paper.

## 2. Dynamic Target Filtering Methods in the 3D Point Cloud

Section 1 of this work highlights how the presence of dynamic objects in realistic scenes can significantly compromise the robustness of SLAM. Unfortunately, many SLAM algorithms based on 3D LiDAR rely on the assumption of static backgrounds and thus are ill-equipped to handle such complex dynamic scenes [46]. Consequently, the point cloud belonging to the dynamic objects must be filtered out to obtain a clean 3D model of the whole scene, as shown in Figure 3. This research falls within the domain of change detection [47], and in this section a comprehensive summary of the universal dynamic point cloud filtering methods based on different principles, and their associated research work will be provided, approaching the topic from the solution-oriented dimension.

### 2.1. Ray-Tracing-Based Approach

This type of method is based on ray tracing [48]. The entire scene space obtained by scanning is initially divided into occupancy voxel grids [49] (or octree grids [50]). The principle of this approach is that the dynamic point cloud only appears in a few scans, so the grid it belongs to is only briefly hit by the laser ray, whereas the laser’s light path passes through it in most cases. The marker is stored in each grid, representing the number of times it is hit or the probability of being hit by laser rays. By traversing all the grids in the point cloud, the grids with marker values below a certain threshold are identified as dynamic grids and all points that fall within them are removed. The illustration in Figure 4 shows the principle of this method. In the last decade, several studies have utilized this approach and obtained promising results.

In [51], Azim and Aycard employed the ray-tracing-based approach for dynamic object filtering in SLAM based on 3D LiDAR. Their proposed method involved identifying dynamic voxels initially by comparing the grid’s status of being marked as free or not free between the before and after scans. Then, by using an algorithm of clustering and by setting a threshold they verified whether these dynamic voxels represented dynamic objects. The study also categorized each cluster by calculating the ratio of the length, width, and height of the enclosing boxes, which further refined the accuracy of dynamic object detection by tracking the motion characteristics of the different object categories. However, the method’s robustness is limited in chaotic dynamic environments, and Azim and Aycard only provided a qualitative analysis of their approach.

Underwood et al. proposed an early and efficient 3D dynamic detection algorithm based on ray-tracing points in a spherical coordinate system [52]. However, the method has a limitation in that it did not construct a global occupancy grid and could only compare two scans at a time. This means that for a map consisting of N frames of scans, it is necessary to perform $\frac{N\left(N-1\right)}{2}$ sub-comparisons to detect all of the dynamic points in the worst-case scenario, which consumes significant computational resources. Moreover, the algorithm requires two parameters: angle and distance thresholds, and the two-dimensional parameter space makes it more challenging to obtain the optimal parameters.

Xiao et al. proposed a method that takes into account the volume of laser rays and evaluates the consistency between points and neighboring rays in different datasets using DST (Dempster Shafer Theory) [53,54]. They incorporated point-to-plane distance information to handle the occlusion problem, but their approach is still at the point level and does not involve voxelization for the dynamic point cloud. Another study by Postica also used DST to evaluate the occupancy of LiDAR scans and identify dynamic points in the point cloud [55]. It proposed a ground plane removal algorithm and an image-based verification step to optimize the problem of false positives, which involves filtering out static points judged as dynamic. Additionally, an octree indexing strategy was used to improve computational efficiency. The method was validated on three short sequences from the KITTI dataset [56]. However, the approach ignores the part of the point cloud beyond the 30m range of the sensor and still requires pre-processing of ground points and the use of cameras, which involves matching images to the point cloud data.

Asvadi et al. proposed a ground estimation method followed by modeling objects above the ground based on grids to distinguish between dynamic and static objects [57,58]. The method used a data structure of voxels to identify dynamic points based on the probability of a voxel being occupied. However, it requires certain assumptions about the environment due to occlusion, and the ground plane features need to be extracted in advance. Similarly, Chen et al. divided the scanned scene into 3D ray voxels and constructed a background model for each voxel [59]. Then, the models were combined by adaptive weighting to detect dynamic points for the removal and restoration of the background. However, the effectiveness of this method for removing dynamic objects other than pedestrians has not been verified. Furthermore, the method only uses the point distance to the scanner as a geometric attribute for modeling, which lacks some accuracy. In another study, Gehrung et al. used the accumulation of probabilistic masses to represent the inherent capability of an object to identify dynamic objects by setting an upper and lower threshold and using background subtraction to remove them [60]. However, this method may easily mark larger static background spots as dynamic objects and is it less robust for special cases.

In contrast to the above methods, Hornung et al. employed an octree data structure to store occupancy information [38], with each node in the octree storing the occupancy probability of that node as log-odds. This method uses the voxel traversal to update the occupancy information of the nodes, where all voxels that are within the sensor’s line of sight but contain a non-empty set of laser endpoints are marked as dynamic. The fully constructed occupancy grid acts as a binary classifier, filtering points from the actual map. However, the probabilistic update function used by this method is very sensitive and it applies to free space where static objects are not particularly abundant.

In [32], which is based on [38], a combination of a trained neural network and an octree grid is used to first obtain a bounding box using an AVOD (Aggregate View Object Detectionnet) work to achieve the initial detection of dynamic objects. The points in the point cloud were then divided into *object points* and *non-object points*. For the *non-object points*, the occupancy rate of all grids on the optical path is reduced and the occupancy rate of endpoints is increased during the raster traversal, whereas for the *object points*, the occupancy rate of all grids and endpoints are reduced. Additionally, for the ground region where the ray-tracing-based approach has a high miss rate (due to the large incidence angle of the laser), this study extracted ground points in the point cloud and maintained a counter in each grid containing ground points, which indicates the number of times that the voxel is classified as ground. Grids with counter values greater than a certain threshold are considered ground voxels and are not marked as free grids during ray traversal. 

In recent research, Zhang et al. introduced a unified benchmarking framework to evaluate dynamic point cloud-removal techniques [61]. They began by analyzing the strengths, weaknesses, and areas for improvement of several widely used non-machine learning methods. Specifically, they examined the algorithmic design and frameworks of *Removert* [34] and *ERASOR* [31], reconstructing their code to eliminate dependencies on ROS for easier benchmarking and faster execution. Additionally, they placed a significant emphasis on optimizing and enhancing the *OctoMap* method [38], which motivated the inclusion of their work in Section 2.1. To mitigate the influence of noise and outlier points in the original *OctoMap* method [38], the researchers employed Statistical Outlier Removal technology for filtering. Subsequently, Sample Consensus segmentation was applied to estimate the ground in the output point cloud. To optimize the process further, they designated the grid cells occupied by the estimated ground points as free space, ensuring that no ray casting occurred in these regions. Their study provided an optimized solution for noise removal and ground segmentation in the original *OctoMap* method [38], preventing the mislabeling of ground points as dynamic and preserving the integrity of the final representation. However, this method still grapples with the inherent challenges of high computational resource requirements and relatively slow execution speeds associated with the ray-tracing-based approach.

As one of the most representative algorithms within the ray-tracing-based approach [33], Schauer et al. proposed the method of storing in the voxel grid the IDs of all scans in which the light path hits the voxel, and by calculating and setting the “maximum safe distance” and eliminating the point clouds of scans with the same IDs in adjacent grids; they optimized the various false positives and false negatives existing in the ray tracing-based approach from multiple perspectives. The specific implementation process and optimization schemes for this algorithm will be analyzed in detail in Section 3 below.

Section 2.1 presents a comprehensive overview of various methods based on the ray-tracing-based approach, including their main idea and problems, as summarized in Table 2. Overall, the ray-tracing-based approach is the most widely used method for filtering dynamic point clouds in a 3D point cloud in recent years, and it has high accuracy but also requires significant computational resources.

### 2.2. Visibility-Based Approach

Given the substantial computational costs associated with the ray-tracing-based approach, several studies have proposed the visibility-based approach as a means of reducing the computational burden. This type of method associates a query point and a map point within an almost identical narrow FOV (Field of View), and if a query point is observed behind a previously acquired point in the map, then the previously acquired point is dynamic. This approach is straightforward and intuitive, as illustrated by the following Figure 5, so it has been employed in various studies on dynamic point cloud filtering. 

Early studies proposing the use of the visibility-based approach to detect dynamic objects focused on iteratively updating the state of map points by successively fusing multiple measurements and checking the visibility between the query scan and the map [40,62]. In [40], the point cloud in the priori map is directly transformed to the query scan coordinate system. A point in the priori map is judged to be dynamic if it is crossed by the optical path of a point in the query scan. Given the difficulty in distinguishing dynamic points based on a single crossing, this method proposed a Bayesian probability-based model that leverages multiple pieces of information, such as the normal vector of a point, the distance between two points, and the category of the previous cycle, to calculate a probability that determines whether a point is dynamic or static. This work also represents a seminal contribution to the field of Long-term SLAM. Ambrus et al. constructed and maintained the static structure of the office space, namely the meta-room [62], through multiple observations and iterative improvement of new data. They employed RGB-D cameras to capture data from the environment, which enabled the accurate reconstruction of static structures by eliminating dynamic elements and incorporating previously occluded objects. One challenge of the visibility-based approach lies in identifying associations between query points and map points. The estimated pose of the LiDAR motion may contain errors that impact the accuracy of these associations, leading to the erroneous removal of static points. Regarding this issue, these methods assumed that the correct pose or registration was provided [62] or used a fixed-size association rule (e.g., within 1°) [40]. However, it is clear that both of these types of assumptions lack some degree of accuracy. 

In [35], dynamic points were identified by selecting two reference scans and verifying whether potential dynamic points in the former reference scan were crossed (visible or not) by the laser beam of the latter reference scan. However, this method also assumed that the poses of the scans were given by SLAM, which would theoretically work better in the cases of denser LiDAR beams. Meanwhile, Qian et al. [63] proposed a dynamic SLAM framework based on LIO-SAM [23], RF-LIO, which utilizes a tightly coupled LiDAR-inertial odometry, adds adaptive multi-resolution range images, removes dynamic objects first for the acquired current LiDAR scan based on the visibility, and then employ scan matching. This approach exhibits good dynamic point cloud rejection capability in highly dynamic environments. Both [35] and [63] belong to the category of online real-time dynamic point cloud filtering (explained in detail in Section 3 below), which provides excellent real-time performance. However, the smaller number of scans selected and referenced can lead to slightly poorer accuracy in dynamic point cloud filtering.

On the other hand, Kim et al. used SLAM techniques to obtain an a priori map of the whole environment and then filtered dynamic target points from the point cloud based on the visibility principle [34]. A key challenge associated with the visibility-based approach is the potential erroneous removal of static points. Such errors can arise from factors such as the nature of the object itself (e.g., ground points with large incidence angles or points situated on the edges of object contours and elongated objects) or the lack of accuracy in the poses obtained via SLAM during the coordinate transformation of query scans and maps. In [34], these false positives are mitigated through the construction of multi-resolution range images. This study is a representative work utilizing the visibility-based approach, and its implementation and optimization will also be analyzed in detail in Section 3.

In recent years, significant progress has been made by integrating the visibility principle with semantic segmentation techniques for point clouds. Chen et al. proposed the LiDAR-MOS framework [64], which enables real-time detection and filtering of dynamic objects. Their approach utilizes three classical point cloud segmentation networks, namely RangeNet++ [65], SalsaNext [66], and MINet, to project the point cloud as distance images and compute residuals between the current and previous scans based on the visibility. These residual images are then combined with the current distance image and used as inputs to a fully convolutional neural network. The network is trained with the binary label to effectively distinguish between moving and stationary objects. Compared to Kim et al.’s method, this approach has the advantage of not relying on the pre-constructed map and operates online using only past LiDAR scans, resulting in improved real-time performance. However, one potential drawback is that it might miss some dynamic objects that are temporarily stationary for a certain period of time.

In Section 2.2, the paper discusses various works based on the visibility-based approach, as summarized in Table 3. The visibility-based approach is found to significantly reduce computational costs compared to the ray-tracing-based approach. However, the method faces limitations such as the challenge of ensuring uniform accuracy of the viewpoint, as well as the difficulty of occlusion and recognizing large dynamic objects in the absence of static objects behind them.

### 2.3. Segmentation-Based Approach

This approach aims to eliminate dynamic objects as a whole by identifying their categories. The idea is simple: if it is possible to accurately identify which points belong to pedestrians, vehicles, bicycles, or others then the dynamic point cloud can be easily filtered out. In recent years, machine learning techniques have been widely used in segmentation-based methods [67,68]. This method uses semantic segmentation based on deep learning to label the class of dynamic objects so that the dynamic point cloud can be eliminated directly with a bounding box, as shown in Figure 6. However, this approach heavily relies on supervised labels and they are vulnerable to labeling errors or dynamic objects of unlabeled classes [69]. 

Early works on the segmentation-based approach did not incorporate machine learning techniques. Petrovskaya et al. modeled vehicles using a 2D bounding box and processed it in 2D representation for application it to 3D data [70]. However, early studies relied on manual model crafting rather than the learning-based approach, and they were insufficient in ensuring the efficiency and accuracy of removing dynamic objects, especially in complex, cluttered scenes with a richness of diverse dynamic objects.

There is also a part of early research using static 3D laser sensors for dynamic object point cloud segmentation. Kaestner et al. proposed a generative object detection algorithm that could extract objects of different sizes and shapes from observation data in an unsupervised manner [71]. In [72], Shackleton et al. used 3D grids for the spatial segmentation of point clouds, followed by point cloud surface matching using “spin images,” Kalman filtering-based tracking-assisted classification, and background subtraction-based background segmentation for object segmentation. Additionally, Anderson-Sprecher proposed using the accessibility analysis-evidence-grid background subtraction method [73], which uses accessibility analysis to remove the internal regions of objects and then performs background subtraction to detect foreground objects. However, these methods are not suitable for mobile 3D laser sensors and only focus on dynamic object segmentation and tracking from laser data, ignoring map updates.

Subsequent point cloud segmentation methods have gradually shifted towards the clustering-based approach. Litomisky first used the RANSAC (Random Sample Consensus) algorithm to eliminate large planes [74], then used VFH (Viewpoint Feature Histogram) to differentiate between static and dynamic clusters, and ultimately identified dynamic points by calculating the Euclidean distance between clusters. Yin leveraged feature matching to identify pixel correspondence between two images [75], detected prospective dynamic objects by performing pixel image subtraction, and then applied the Euclidean clustering method to extract clusters from dynamic objects as seeds. Yoon et al. proposed a region growth-based strategy to obtain dynamic clustering from confirmed dynamic points [35]. Despite the usefulness of clustering methods, they are still prone to omitting outlier dynamic points.

As relevant technologies continue to evolve, more and more studies are utilizing 3D point cloud data obtained from LiDAR scans as input for neural networks that output specific detection results to identify dynamic objects. Chen et al. proposed a method for dynamic object detection based on a multi-view 3D network that employs two sub-networks: one for 3D candidate box generation and the other for multi-view feature fusion, thus achieving the goal of 3D object reconstruction [76]. Zhou proposed a voxel encoding method for extracting features from point cloud data [77]. By encoding the point cloud into a voxel array and using 3D CNN(Convolutional Neural Network) for feature extraction, the study achieved an end-to-end learning process for 3D object detection based on point clouds. Similarly, Shi segmented point clouds using the PointNet++ network to generate a small number of high-quality 3D object candidates [78]. The RPN(Region Proposal Network) is utilized to generate reliable 3D object proposals for multiple classes. Using these proposals, the second stage detection network performs accurate orientation regression to predict the range, orientation, and class of objects in 3D space. Ruchti proposed using a neural network to predict the probability of a 3D laser point reflecting a dynamic object [79], determine whether each point belongs to a dynamic object by the prediction result, remove the points classified as dynamic objects, and then give only the remaining static points to SLAM for mapping and localization work. However, the disadvantage of neural network-based dynamic object removal methods is that they cannot detect untrained objects, and they occasionally fail to detect objects.

While many previous approaches were based on detecting actual motion, recent advancements in deep learning have enabled a shift toward appearance-based methods for dynamic object detection. In urban environments, it is often assumed that pedestrians, cyclists, and cars are the most common types of dynamic objects, and these classes can be detected in point clouds using a deep learning-based semantic segmentation approach. As reported in [80], a new SURF(Speeded Up Robust Features) algorithm was proposed that combines convolutional neural networks to extract semantic information from LiDAR 3D measurement point clouds to build maps with labeled surface elements that contain semantics. The algorithm employs a filtering algorithm for dynamic object removal, retaining static environmental information in the built semantic maps. Zhao et al. developed a semantic segmentation network using a dense matrix coding method to convert sparse [81], irregularly scanned LiDAR point clouds into a dense matrix. In [65], a rotating LiDAR sensor model was used to map single-frame point cloud data to the sensor coordinate system, resulting in a continuous range image. A CNN was then used to combine the range images for semantic segmentation to achieve recognition and segmentation in the point cloud. Wu et al. proposed an end-to-end pipeline, named SqueezeSeg [82,83], which transforms LiDAR point clouds into an image format and feeds them into a CNN. The CNN outputs a pixel-labeled map, which is then refined by a CRF(Conditional Random Field) recursive layer. Finally, a conventional clustering algorithm is utilized to obtain instance-level labels. Similarly, Biasutti proposed an end-to-end network for the semantic segmentation of 3D LiDAR point clouds based on U-Net [84]. This network extracts higher-order features of 3D neighbor points for each point and the learned features are then used to perform semantic segmentation using a U-Net segmentation neural network. Additionally, Cortinhal proposed the SalsaNext method [66], which uses a new context module and a residual dilated convolution structure to enhance segmentation accuracy by adding perceptual field and pixel rearrangement layers. However, these supervised learning approaches require manual labeling of training data, which can be costly and challenging to obtain. Moreover, they are limited to the subset of object types that exist in available datasets [10,85,86,87], thus hindering the ability to detect new dynamic objects.

The semantic segmentation of point cloud data based on appearance can identify potentially dynamic objects. However, this method does not necessarily indicate whether the objects are moving or stationary. For instance, an identified vehicle could be parked or in motion. Depending on the different needs for 3D point cloud models, it may be necessary to either remove parked vehicles or retain them. However, pure point cloud semantic segmentation methods tend to prioritize dynamic point cloud removal without distinction in the present moment, so they are combined with some traditional methods at times. Yu et al. combine semantic segmentation network and movement consistency checking to select moving and immobile points [46]. This method calculates the movement consistency of feature points and uses the proportion of moving points in the same object to determine whether the object is moving or stationary. Finally, dense semantic octo-tree maps are constructed using methods such as crawling algorithms. Sun et al. express the map in the form of an OctoMap and model each cell as an RNN(Recurrent Neural Network) to implement a Recurrent-OctoMap [88]. To extend the observation period of the Recurrent-OctoMap, the approach also develops a robust 3D localization and mapping system for continuously mapping dynamic environments. Meanwhile, Chen et al. combined the visibility-based approach with point cloud segmentation, the range images and residual images are used to train a CNN [64], producing a binarized label for each measurement indicating whether it belongs to a moving or static object. The specific content of their method is introduced in Section 2.2. 

In recent work, Mersch et al. introduced an innovative concept involving the application of sparse 4D CNN to extract spatio-temporal features and to segment points into moving and non-moving [89]. To enhance the precision and recall of moving object segmentation, they employed a voxelized binary Bayesian filter, which provides a probabilistic representation of dynamic environments. He et al. introduced an end-to-end sparse tensor-based CNN [90]. Their method leverages the AR-SI theory [91] to create an autoregressive system-identification filter that discerns moving objects in a 3D point cloud, providing temporal features for CNN inputs. Subsequently, the temporal features are combined with special features extracted from the original point cloud. By mapping the labels to segmentation training, the method produces accurate IoU results. Toyungyernsub et al. proposed an integrated framework for static-dynamic object segmentation and local environment prediction [92]. This framework utilizes an occupancy-based environmental representation and incorporates modules for static-dynamic object segmentation and environment prediction. The static-dynamic object segmentation module employs the SalsaNext [66] neural network architecture to detect static and moving objects within the environment. The environment prediction module utilizes the predicted static-dynamic segmentation output as input to forecast the future occupancy state of the environment. It is worth noting that, despite the promising results yielded by [46,64,88], these methods still require manually labeled, high-quality datasets. 

In Section 2.3, this paper delves into the implementation of various algorithms based on the segmentation-based approach and explores their strengths and limitations, which are summarized in Table 4. In general, deep learning-based semantic segmentation methods can effectively and efficiently detect dynamic targets in point clouds and filter them out. However, one major limitation is that they can only recognize pre-trained dynamic object types, and other types of dynamic objects cannot be detected. Additionally, the requirement of GPU resources for deep learning-based methods results in a higher cost.

### 2.4. Other Approaches

There exist several other methods for filtering out dynamic point clouds in a 3D point cloud, which are based on specific assumptions and have achieved good results in certain scenarios. In this section, these methods will be introduced. However, it is important to note that these methods are not systematic and rely on certain conditions and constraints.

In [93], an approach for joint self-localization and generic object tracking is proposed, which integrates object detection and tracking with sensor trajectory estimation. The method involves pre-processing the collected 3D point cloud, computing features, and generating object hypotheses, followed by generating a trajectory for each hypothesis and tracking them over time. Trajectories that exhibit consistency are labeled as dynamic by the learned classifier. However, a limitation of this method is that it assumes that objects can be tracked continuously in subsequent scans, which may not hold for some dynamic objects in real scenes, such as pedestrians.

An alternative approach was proposed by Dewan’s team [94,95], who used rigid scene flow, i.e., point-wise velocity estimation, to detect dynamic objects. They compared subsequent point clouds and identified motion sequentially using a voting scheme. The first detected motion will always be the relative motion of the stationary environment, followed by the largest dynamic objects. Motion cues were used for the segmentation and tracking of moving objects, and point-level matching was performed between successive scans. Their work was compared with that of [93] and was shown to have superior performance. However, the method requires prior rejection of ground points in the pre-processing step, and relies on the minimum velocity assumption, making it difficult to distinguish the scene stream from the noise for slow-moving dynamic objects. Therefore, it is also unsuitable for filtering out pedestrians.

In [96], a scan-to-model-based matching framework is proposed that uses the localized LiDAR scan of the previous region as a model with an IMLS(Implicit Moving Least Squares) surface representation, which is known for its good approximation and smoothness, and for its superior noise and outlier reduction. The method has demonstrated good performance in KITTI benchmark tests with excellent low-drift characteristics. However, the method assumes that all objects smaller than a certain size are potentially dynamic objects, which may result in the loss of valuable features for matching or 3D reconstruction.

Ushani et al. proposed a learning-based method to compute scene streams that use 3D LiDAR sensor data to construct the occupancy grid [97], extract foregrounds through a learned background filter, and use the filtered occupancy grid to compute the original scene stream between successive scans to roughly identify dynamic point cloud, which is then refined by a learned classifier. However, as the method assumes that the motion of dynamic objects is restricted to a horizontal plane and only constructs the occupancy grid of 50 m × 50 m, it is clearly not suitable for dynamic detection over large areas.

Lim et al. proposed an innovative method for detecting and filtering a dynamic point cloud in a 3D point cloud based on the prior assumption that the majority of dynamic objects might be in contact with the ground [31]. Similar to the visibility-based approach, this work identifies dynamic objects by comparing the difference between query scan and submap. However, this method divides the scanned spatial region into bins and compares the height difference of the point cloud in the z-direction within each bin. If the ratio of the extreme height difference within a bin is below a certain threshold, it is inferred that dynamic objects are present, and all points above the ground points in this bin are removed. This method has overcome many limitations of the ray-tracing-based approach [33] and the visibility-based approach [34], as discussed in detail in Section 3. However, it assumes that dynamic objects are in contact with the ground, and it may encounter a higher risk of failure in more complex dynamic environments.

Wang et al. also proposed an approach to distinguish between dynamic and static points using the vertical voxel height descriptor [98], but the key distinction is that their method does not rely on a complete global map and enables online real-time processing of dynamic point cloud. In their system, akin to RF-LIO [63], the IMU data is pre-integrated to obtain initial attitude outputs, then the pre-processing module involves feature extraction and segmentation of scans, and local map construction. The dynamics removal module employs the vertical voxel occupancy descriptor to filter out dynamic objects and performs matching between scans and the submap to achieve accurate attitude estimates. However, this kind of method that relies on vertical descriptors to identify dynamic objects inevitably has inherent limitations, leading to compromised localization accuracy and robustness when dealing with sloping pavements.

In Section 2.4, this study highlights the implementation ideas and problems of certain methods for filtering out a dynamic point cloud. These methods are summarized in Table 5. While these methods can provide better results than traditional approaches in certain scenarios, they are often based on subjective assumptions and lack sufficient robustness in complex and chaotic dynamic environments.

## 3. Processing Strategies of Dynamic Objects in the SLAM Framework

Section 2 introduces different methods for filtering dynamic targets in a 3D point cloud in the *solution-oriented* dimension. In this section, this paper explores the *problem-oriented* dimension and investigates different processing strategies that correspond to different categories of dynamic objects in the SLAM framework. Initially, an overview of the SLAM system framework had been provided before objects have been classified into four categories based on their dynamic characteristics: high dynamic objects, low dynamic objects, semi-dynamic objects, and static objects. For different categories of dynamic objects, three distinct processing strategies have been summarized: *Online real-time filtering*, *Post-processing after the mapping*, and *Long-term SLAM*. The paper also performs a detailed analysis of different methods within the same processing strategy.

### 3.1. The Overall Framework of the SLAM System

SLAM technology has made significant strides over the past few decades. As far back as 1992, the feature-based fusion SLAM framework [99] was established. The SLAM framework based on 3D LiDAR primarily organizes the SLAM according to its functionality, namely, localization and mapping, as shown in Figure 7. As SLAM technology has developed, localization and mapping have evolved from two distinct modules into a comprehensive system. The two modules promote each other. The high-precision odometer composed of multiple sensors provides real-time pose estimation for the robot and the basis for the reconstruction and stitching of the 3D scene. Similarly, high-precision 3D reconstruction provides important data for pose estimation for feature-based odometry. Even a separate odometer system is also inseparable from the establishment or storage of temporary local maps to assist pose estimation [100].

In terms of components, the modern SLAM system based on 3D LiDAR can generally be divided into two stages: the front-end and the back-end, as illustrated in Figure 8. The front-end is responsible for the real-time collection of raw data from the environment, feature extraction and matching, pose estimation for the current scan, and storage of corresponding local map information. Meanwhile, the back-end is primarily responsible for loop closure detection, triggering global drift correction, and optimizing pose estimation for large-scale 3D scene reconstruction. Precisely because the front-end and back-end belong to different stages with different roles and receive and process different data and information, different processing strategies exist in the SLAM framework based on 3D LiDAR for different classes of dynamic objects.

### 3.2. Classification of Objects Based on the Dynamic Degree

In the SLAM framework, the LiDAR scans point clouds of various objects, most of which are permanent static objects that need to be retained for subsequent feature matching, pose estimation, and mapping, but there are also many dynamic objects that, due to their different degrees of dynamics, have different processing strategies. For instance, while vehicles are dynamic objects, vehicles driving on the road, waiting for traffic lights, and parked in parking lots are clearly separate categories of dynamic objects requiring different processing strategies in the SLAM framework. In certain studies, dynamic objects are often categorized into two broad groups: high dynamic (HD) and low dynamic (LD) [101]. However, this classification may lack precision, resulting in a potential compromise on accuracy and efficiency during the subsequent filtering process. This paper proposes a more refined classification of objects based on the degree of their dynamics, dividing them into the following four categories:**High dynamic objects**: objects that move continuously in the LiDAR scan, such as people walking on the street, moving vehicles, and running pets.**Low dynamic objects**: objects that are in a transient state, such as people standing on the street talking, and vehicles stopped waiting for a traffic light.**Semi-dynamic objects**: objects that remain stationary during a SLAM cycle, but not forever, such as vehicles in the parking lot, stacked materials, temporary sheds, temporary fences, temporary stages, and mobile catering vehicles on the roadside.**Static objects**: objects that are permanently immobile, such as walls, buildings, and structures, roads, traffic signals, and the vast majority of fixed facilities.

The first three types of objects can be collectively referred to as dynamic objects, and for different categories, different stages of processing strategies can be used in the SLAM framework. For example, high dynamic objects can be filtered out online in real-time, while low dynamic objects may be rejected using post-processing after the mapping. Meanwhile, semi-dynamic objects necessitate the use of Long-term SLAM to detect changes between different sessions of SLAM. It is worth noting that these four types of strategies are upwardly compatible. For instance, Long-term SLAM can also filter out high and low dynamic objects with greater accuracy, but the real-time performance decreases due to the increased information required for reference and comparison.

### 3.3. Online Real-Time Filtering out of Dynamic Objects

This strategy focuses on the online filtering of dynamic objects during SLAM based on 3D LiDAR, ensuring real-time availability of a static model of the entire scene while the SLAM is being completed. It offers the advantage of excellent synchronization and real-time performance. However, due to the limited amount of data that can be referenced and compared, which are typically only adjacent scans from a short period of time before and after the current scan or local submaps comprising these recent scans, the effectiveness of the strategy in rejecting dynamic objects is limited. Consequently, this strategy is primarily suitable for filtering high dynamic objects in the scene.

As mentioned in Section 3.1, during the LiDAR-based SLAM framework, the front-end is responsible for aligning point clouds between different scans, while one of the functions of the back-end is to perform the construction of the overall 3D model of the scene. The strategy of *online real-time filtering out of dynamic objects* can be implemented either in the front-end during the registration or in the back-end during the mapping.

Currently, mainstream point cloud alignment methods are based on the assumption of a static background. Ideally, it is best to filter out dynamic objects before or during the front-end alignment to ensure that the final result is based on static point clouds. In SLAM based on 3D LiDAR, research on rejecting dynamic objects during front-end alignment is relatively limited. The approaches are generally divided into two types: traditional methods and deep learning. Traditional methods often rely on inter-scan comparison to separate dynamic points, such as eliminating points that are too far away or comparing differences between the current scan and submap. In contrast, deep learning-based methods can directly detect dynamic objects based on their categories from the point cloud and remove them before alignment, which is faster and more accurate.

Qian et al. proposed a method based on the visibility-based approach, RF-LIO [63], which filters out dynamic points prior to scan matching. Their method builds upon the LIO-SAM [23] framework, with the addition of range image construction and dynamic point rejection. Upon a new scan arrival, an initial position estimate is obtained using IMU odometry, and an initial resolution is determined. This resolution is then used to construct the current scan and the corresponding submap as range images, which are compared based on visibility to detect the dynamic points in the submap for removal. RF-LIO is a traditional method for rejecting dynamic points at the front-end of SLAM, but due to the limited number of scans available for comparison, it can only be used to filter out high dynamic objects, and its application scenario is limited by the shortcomings of the visibility-based approach.

Pfreundschuh et al. present a deep learning-based approach to reject dynamic point clouds in the front-end [102], which is gaining popularity compared to conventional methods. The method combines traditional dynamic point cloud filtering methods with deep learning point cloud segmentation. It starts by constructing an automatic tagging pipeline based on the concept of ray-tracing-based dynamic point cloud detection, which is used to generate a dataset for training and testing. Then, 3D-MiniNet is trained with this dataset to obtain an end-to-end network, which directly segments the dynamic objects in the original point cloud. Finally, the clean point cloud after filtering is provided to LOAM [20] for subsequent SLAM works. This approach presents an unsupervised generic dynamic object-aware LiDAR SLAM method, which significantly reduces the cost of manual tagging in the dataset and human interference and improves the accuracy of dynamic object rejection. However, a common issue with learning-based methods is that dynamic objects that belong to untrained classes cannot be recognized.

In most real-life scenes, dynamic objects usually do not cause significant interference during point cloud alignment, but their presence can lead to the ghost-trail effect that can greatly impact the mapping stage. So it is more common to perform online real-time filtering out of dynamic objects during the mapping in the back-end. Various methods are used to handle dynamic points in key frames inserted into the map in real-time, resulting in a clean 3D scene map that can be obtained while the mapping is completed. Several works have utilized the real-time filtering of dynamic objects during the mapping of SLAM, including [51,53,54,55,94,95], among others. Two studies employing this strategy are presented below.

In [35], a concise idea was proposed where one scan is taken as a reference frame before and after the query scan. The front reference scan is used to compare the point-to-point distance, and the points with too large a distance are identified as potential dynamic points, which are then put into the back reference scan for verification, and if they are crossed by the laser beam of the back reference scan, they are confirmed as dynamic points. The confirmed dynamic points are used as seeds to grow clusters and obtain dynamic clusters. The whole process of dynamic point cloud filtering does not require constructing local maps or submaps, only two reference scans.

Fan et al. present a recent study that uses the real-time filtering out of dynamic objects strategy [103]. The method starts with an initial screening of dynamic targets using the visibility-based approach to enhance the efficiency of subsequent computations. The static submap M_S_ obtained after initial screening is then optimized online during back-end mapping to obtain accurate dynamic point cloud filtering results. The core idea of the back-end algorithm is based on the ray-tracing-based approach. The voxels are marked as free when light passes through them, and a visibility check is introduced to approximate the denominator *nfree* of the occupancy probability, which is the number of light rays passing through each voxel. This process speeds up the ray-tracing process and greatly reduces computational resource consumption.

In Section 3.3, the paper presents the first strategy for processing dynamic objects in SLAM based on 3D LiDAR: *Online real-time filtering out of dynamic objects*. The relevant methods are summarized in Table 6. This strategy has the advantage of being synchronized with SLAM, which has excellent real-time performance without requiring additional time. However, the downside is that in order to achieve real-time filtering, only a limited number of reference scans are used, which may result in significant differences between subsequent point cloud comparisons due to occlusion or sparse data. As a result, this strategy is only suitable if filtering accuracy is not highly demanding, and it is only applicable to high dynamic objects.

### 3.4. Post-Processing of Dynamic Objects after the Mapping

This strategy involves post-processing to filter out dynamic objects after the completion of the SLAM mapping, in conjunction with the overall map of the 3D scene. It can refer to all the scanned point clouds in the entire SLAM cycle, making it more conducive to the detection of temporarily immobile objects (i.e., low dynamic objects) and of course, the identification of high dynamic objects is even easier. The post-processing strategy can achieve more accurate and thorough filtering of dynamic objects, but the drawback is that the dynamic objects filtering lags after the SLAM mapping, resulting in a further time delay after SLAM before the final static map is available.

Numerous studies have adopted the strategy of post-processing after mapping because it considers the information of all global scans and enables accurate and thorough dynamic filtering. Early works include [38,40,52,104,105], while recent works include [31,32,33,34,106,107], and others. Despite using the same post-processing strategy, these studies have employed different methods for filtering dynamic point clouds. The paper has selected three different methods that were applying this strategy, providing detailed analyses of their algorithmic principles, implementation process and optimization ideas.

The *peopleremover* method [33] is a post-processing filtering of dynamic point cloud method using the ray-tracing-based approach, whose main contribution is to provide effective optimization for the false positives (i.e., static points are incorrectly identified as dynamic points for removal) and false negatives (dynamic points in the static grids are incorrectly retained). The concept of the proposed algorithm lies in the storage of information in each voxel, which is the ID of all scans that have hit that voxel rather than the number or probability. This optimization significantly reduces the amount of data to be stored. The ray-tracing-based approach is prone to producing false positives in certain local special cases, such as ground points or corner points. This is due to challenges such as excessively large angles of incidence or occlusion. To overcome these, this work proposes a “maximum safe distance” approach. For a given laser point, traversal of all grids on its optical path is halted prematurely once it reaches a specified distance from the point, ensuring that the static grids near the point are not mistakenly marked as dynamic. For false negatives as shown in Figure 9, the left image displays point cloud-obtained scans of a person standing on the ground, consisting of person points and ground points. Filtering of the dynamic points produces a point cloud as depicted in the middle image, where the three red points situated above the ground are evidently dynamic (i.e., person points) but are erroneously retained. To address this issue, this work proposes an approach. During the process of marking the dynamic grid, the IDs of scans that hit it are examined in the adjacent voxels, and if present, the point clouds of the scans corresponding to these IDs in the adjacent voxels are removed. In the right image, the ID of the scan that generates red points is stored in the B2 and C2 grids, so that the red points in the adjacent B1 and C1 grids are also filtered out.

The *Remove, then Revert* approach [34] proposes the post-processing strategy based on visibility to filter out dynamic point clouds. This method adopts the multi-resolution range image to address the false-positive problem that is commonly encountered in the visibility-based approach. The optimization ideas presented in this study have been referenced in subsequent studies. A set of original neighboring scans and their corresponding transformed poses to the global map (SE(3)) are given. A scan from this set is designated as the query scan, denoted as P^Q^, while all other scans are consolidated into a submap, denoted as P^M^. P^Q^ and P^M^ are converted to the same coordinate system and then projected as range images. If a point in P^M^ is shallower in depth than its corresponding point in P^Q^, it is identified as a dynamic point and eliminated. The points in P^M^ can be divided into two categories: the dynamic point set D_1_ and the static point set S_1_, and D_1_ contains both true and false positives. The study found that using range images of lower angular resolution resulted in a significant reduction in false positives, and it takes advantage of this property to propose a two-step method. First, the most rigorous dynamic point deletion is performed using the highest resolution range images, and this yields a preliminary dynamic point set and a static point set. Then, the image resolution is reduced and re-matched to revert the points that were previously considered dynamic but are now considered static points to the static point set. This process is repeated for all scans and resolutions until the final static map is obtained. The pipeline of this algorithm is depicted in Figure 10:

*ERASOR* [31] is a method of post-processing for filtering out dynamic point clouds based on dynamic objects in contact with the ground, effectively overcoming the limitations of the ray-tracing-based and visibility-based approaches. This work considers that most dynamic objects make contact with the ground, so if the points are present above ground points in P^M^ but not in P^Q^ at the same location then they can be classified as dynamic objects. The specific method involves the creation of a spatial region in both the P^Q^ and P^M^, referred to as the *Volume of Interest* (VOI). This region is then divided into bins based on specific angles and radii, and the differences between the point clouds above the ground points in the P^Q^ and P^M^ are then compared within each bin. This method introduces a descriptor called the *Region-wise Pseudo Occupancy Descriptor* (R-POD), which is utilized to represent the height difference in the z-direction of the point cloud in each bin. Moreover, the work proposes an operation known as the *Scan Ratio Test* (SRT), which identifies possible dynamic regions by comparing the difference in descriptors calculated for bins at the same location in the P^Q^ and P^M^. The operation calculates the ratio of the two descriptors, and the bin with a ratio less than the threshold value is identified as a potential dynamic region while other regions are preserved. The final step is to fit ground points within all the bins marked as dynamic regions, and then remove all points lying above the ground points as dynamic points. The ground extraction method used was R-GPF(Region-wise Ground Plane Fitting). The eigenvalues and eigenvectors of the ground points were calculated using PCA(Principal Component Analysis), then the plane equation of the ground was calculated and the points above the ground were removed as dynamic points. Finally, dynamic bins that had been processed were combined with the static bins that were preserved previously, resulting in a global static map. An overview of this work is illustrated in Figure 11:

In Section 3.4, the paper presents the main ideas of the post-processing strategy and of several different methods that employ it. The advantages and disadvantages of each method are also summarized in Table 7. Overall, the post-processing strategy is the most widely used tactic because it does not require real-time considerations and can use all scans in the entire SLAM cycle as reference information to identify dynamic objects, which is more accurate for dynamic point cloud filtering. The post-processing strategy can filter out both high and low dynamic objects, making it ideal if high dynamic object rejection accuracy is required. However, the proposed strategy may not be applicable to scenes with a very high percentage of dynamic objects, as it becomes challenging to obtain accurate poses between each scan and the global map in such scenarios. In addition, the specific method used to filter out dynamic objects in this strategy needs to be carefully selected based on the scene characteristics.

### 3.5. Long-Term SLAM

The two strategies presented above occur at different stages within a SLAM cycle, but semi-dynamic objects are typically stationary and undetectable throughout a single SLAM cycle. However, in many application scenarios, it is necessary to maintain a long-term model of the environment, which requires filtering out these semi-dynamic objects. *Long-term SLAM* is a strategy that detects and removes changed objects between different SLAM cycles while updating and maintaining the environment map. As an emerging field of SLAM research, the measures used to detect and filter out dynamic objects in *Long-term SLAM* remain similar to those described in Section 2. The uniqueness of this strategy is in the temporal and spatial alignment and fusion of trajectories and maps from different SLAM cycles, as well as the updating and maintenance of the long-term 3D scene map.

Pomerleau et al. presented seminal research in the field of *Long-term SLAM* [40]. The authors argue that the core of map fusion in *Long-term SLAM* lies in how to handle the parts of the map that change between different sessions, i.e., high and low dynamic objects and semi-dynamic objects. Their dynamic object detection and filtering method is based on visibility, but instead of using range images, they transform the point cloud in the map directly into the query scan coordinate system. Dynamic points are initially filtered based on whether the points in the map are crossed by the light paths of the points in the query scan, and information from multiple sources is fused to further identify dynamic objects in the map that need updating. This work is significant as it unifies dynamic object filtering and map updating in LiDAR-based SLAM, serving as an important guideline for future *Long-term SLAM* research. However, the authors’ assumption of precise poses between each query scan and map may not hold in real outdoor environments.

In previous years, several *Long-term SLAM* studies have proposed using different map representations such as NDT(Normal Distribution Transformation) maps [108], occupancy grid maps [14,88,109], or frequency domain representation [110]. Einhorn presents a generic SLAM method that can be used for both 2D and 3D scenes [108]. It combines the occupancy grid map with the NDT map and modifies the NDT mapping method to better handle free-space measurements and dynamic object detection. This method is both a SLAM mapping algorithm and supports localization and map updates. Another study proposes an online update method for 2D grid maps that uses a graph-based SLAM system to anchor local maps [109], then combines occupancy estimates from multiple local maps to produce a global occupancy map. To handle too many local maps to maintain the quality of the global rendered map, the employed method prunes redundant local maps using algorithms that determine their cost and contributions. However, for change detection in 3D environments, dealing directly with a raw 3D point cloud may be preferred over dealing with other forms of maps.

One of the most recent studies using Long-term SLAM is presented in [4,101]. The former introduces the concept of sessions, where the first map build belongs to session zero, and an a priori map of the current environment is constructed and stored in the database. When the robot performs a task, it enters localization mode, where each localization reads the map data from the database and creates a new session. The old submap is continuously pruned to limit its number, and the new submap is added to the global map of the current session. When the old submap is deleted, sparsification of the pose optimization is triggered. Finally, at the end of the localization task, the updated map is saved and passed into the database, thus completing a map update. As can be seen, the focus of this study is not on detecting dynamic objects during *Long-term SLAM*, but rather on updating the map in real-time to guide the current motion of the robot, which has good applications in highly dynamic places such as supermarkets and shopping malls.

In [101], the focus lies on map building and management in the context of *Long-term SLAM*, which involves detecting differences in SLAM between different sessions at different times and updating these changes to the central map. The authors propose a *Long-term SLAM* system, called “*LT-mapper*”, which comprises three sub-modules: *LT-SLAM, LT-removert*, and *LT-map*. The LT-SLAM module tackles the problem of trajectory alignment between multi-temporal SLAM by using the anchor node-based loop back factor, and ICP iterative computation is used to obtain the relative constraints between the corresponding keyframes of two different sessions to achieve the alignment between trajectories. The *LT-removert* module, based on the *“Remove, then Revert”* method, initially filters the dynamic objects HD (i.e., high and low dynamic objects) in a single session. For the dynamic objects LD (i.e., semi-dynamic objects) between different SLAM sessions, the kd-tree is constructed in the central map, and the neighborhood search is performed in the target point cloud in the query map. The “*Remove, then Revert*” method is again used for false positives in LD to further revert them to static points. Finally, the *LT-map* module’s task is to extract the HD and LD (i.e., all dynamic objects) from the change detection to form delta maps and to update the central map by uploading delta maps. By solving the problem of temporal and spatial alignment and integration of the trajectories and maps of different sessions of SLAM, *LT-mapper* effectively maintains a long-term 3D scene map.

In Section 3.5, the paper presents *Long-term SLAM* as the ultimate strategy for dynamic object filtering in SLAM based on 3D LiDAR, and the related works are summarized in Table 8. While *Long-term SLAM* is a more comprehensive and systematic approach than the other two strategies, its goals are not limited to detecting dynamic objects but rather encompass a range of core tasks such as long-term map updating and maintenance, as well as real-time robot localization, motion guidance, and map correction.

In Section 3, the paper categorizes objects based on their dynamic degree and summarizes three corresponding strategies for different classes of dynamic objects at different stages of SLAM based on 3D LiDAR. Table 9 summarizes the applicability and compares the pros and cons of the three strategies. It should be noted that the division of the different categories of strategy is problem-oriented, and the specific measures used in each are those dynamic point cloud filtering methods mentioned in Section 2: ray-tracing-based filtering, visibility-based filtering, segmentation-based filtering, and other approaches. Moreover, the algorithm principles of the corresponding methods selected for different strategies may be the same.

## 4. Conclusions and Future Outlook

Over the years, SLAM based on 3D LiDAR has made significant progress. However, most of the mainstream algorithms assume a static background, which is not the case in real-world scenes that are filled with various dynamic objects. These dynamic objects can cause interference and impact the front-end and back-end operations of SLAM, resulting in serious damage to its accuracy and robustness. Therefore, it is necessary to filter out the dynamic point cloud in SLAM based on 3D LiDAR and to obtain a clean 3D scene model.

This paper aims to classify and summarize various universal methods for filtering out dynamic point clouds in the 3D point cloud and different strategies for handling dynamic objects in SLAM based on 3D LiDAR. The dynamic point cloud filtering methods are solution-oriented, while the three processing strategies are problem-oriented, serving the practical applications of dynamic point cloud-removal technology within the realm of SLAM based on 3D LiDAR. From a broader perspective, all of these strategies, methods, and algorithms can be summarized under one keyword: change detection. Detecting changes between a scan and several scans, or between scans and the map is the key to filtering out high and low dynamic objects, whereas semi-dynamic object updating is the change detection between scans and maps of different SLAM cycles. This paper discusses how change detection is performed during SLAM based on 3D LiDAR and analyses the advantages and disadvantages of each study. However, this work is not exhaustive, and only a few representative studies are included to provide readers with a general understanding and a starting point for tackling similar problems.

In future research, dynamic point cloud filtering methods based on deep learning are poised to be a significant development direction. One proposal by [102] is the use of deep learning methods to train and generate datasets, and as related technologies advance, future object training will cover various categories in realistic scenes to achieve accurate identification of dynamic objects in the original point cloud at the front-end. This will provide SLAM with clean point clouds for alignment and subsequent works, thus improving and extending the performance and functionality of SLAM. *Long-term SLAM* is also a promising area for future research. The ultimate goal of SLAM is to enable autonomous robot operation or autonomous vehicle driving. Achieving accurate localization and operation in a dynamic environment that changes constantly, while judging the differences between the current environment and the priori map and correcting the map in real-time, will require Long-term SLAM. Further research and development of Long-term SLAM technology will be necessary to achieve this goal.

## Figures and Tables

**Figure 1 sensors-24-00645-f001:**
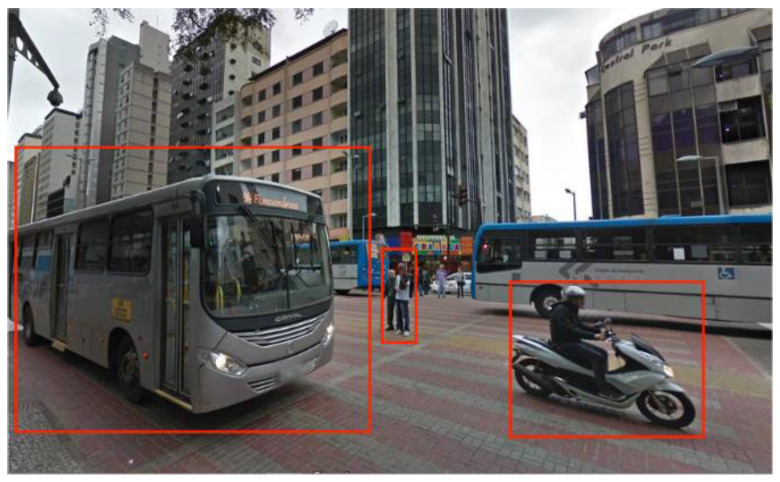
Common Dynamic Objects in the street view.

**Figure 3 sensors-24-00645-f003:**
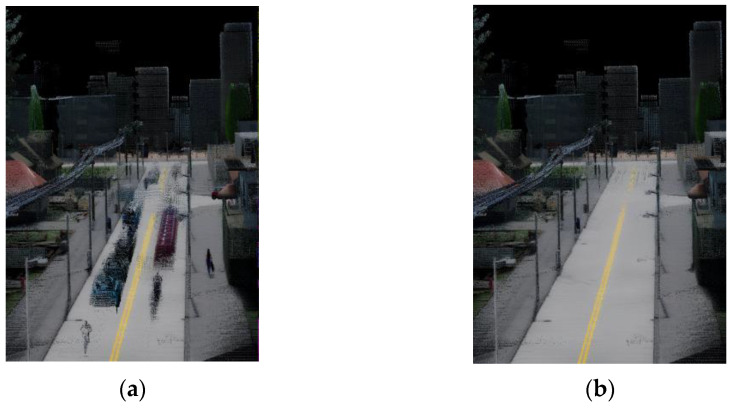
(**a**) The original unprocessed point cloud; (**b**) The point cloud after filtering out dynamic target points.

**Figure 4 sensors-24-00645-f004:**
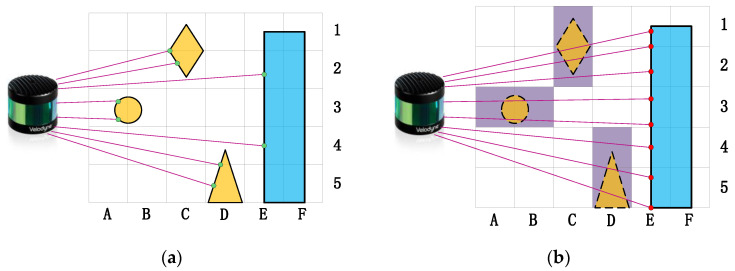
The gray raster marks the 2D voxel boundaries. Magenta lines mark the scanner lines of sight. The blue shape represents a static object, while the yellow shapes represent dynamic objects. Green and red dots represent the measured points taken from two different scans: scan-I and scan-II. (**a**) The dynamic objects appear briefly in scan-I and are hit by lasers. (**b**) The dynamic objects disappear in scan-II and are passed through by lasers, and the grids A3, B3, C1, C2, D4, and D5 (the purple grids) where they are located are identified as dynamic grids, allowing filtering out of the point cloud within them.

**Figure 5 sensors-24-00645-f005:**
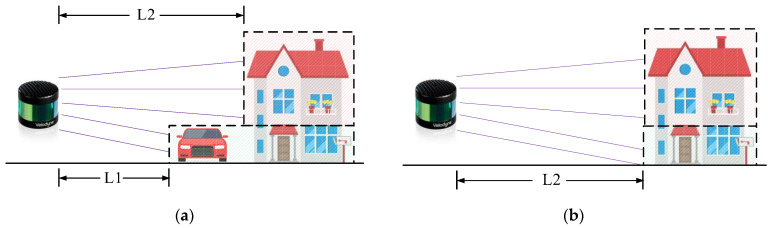
(**a**) When the vehicle drives past the building and blocks the view of the LiDAR, the nearest distance of the point cloud in the red area from the sensor is L2, while the nearest distance of the point cloud in the green area from the sensor is L1. (**b**) After the vehicle moves away, the closest distance between the sensor and the point cloud in the green area is restored to L2, thereby indicating the presence of the building point cloud behind the car point cloud. Based on this, the car can be identified as a dynamic object.

**Figure 6 sensors-24-00645-f006:**
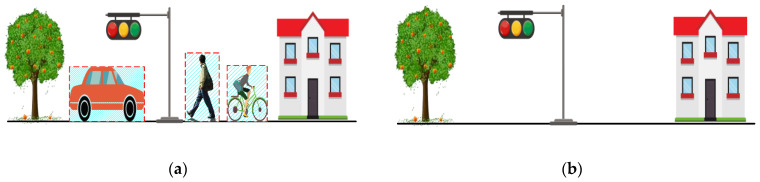
(**a**) Dynamic objects are accurately identified for their respective categories, such as vehicles, pedestrians, or cyclists, and the detection bounding box is formed around them to segment the dynamic point cloud; (**b**) the point clouds inside the bounding box are removed to obtain a static scene model.

**Figure 7 sensors-24-00645-f007:**
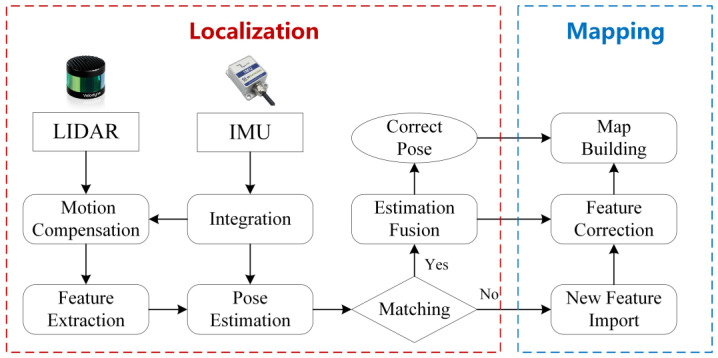
The basic framework of SLAM based on 3D LiDAR.

**Figure 8 sensors-24-00645-f008:**
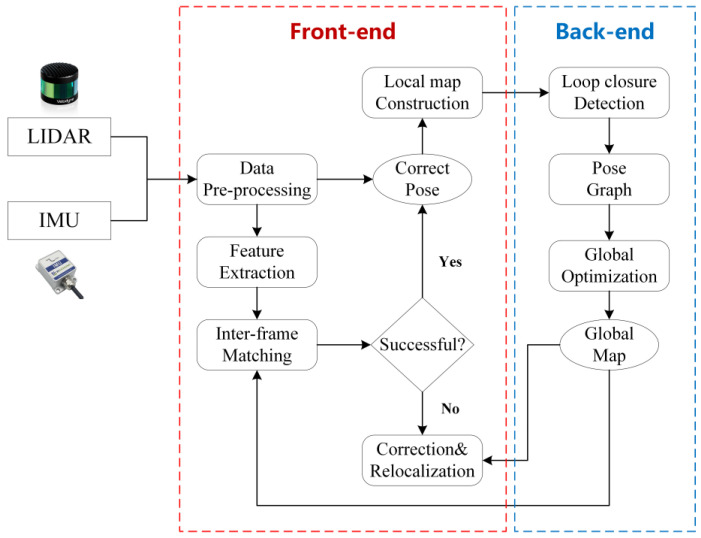
The main components of the SLAM system based on 3D LiDAR.

**Figure 9 sensors-24-00645-f009:**
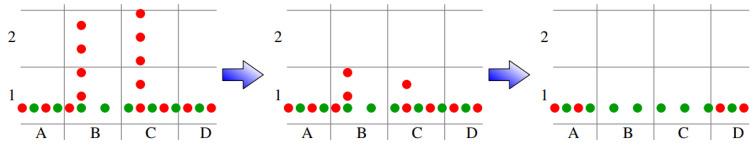
Optimization of false negatives in the peopleremover method.

**Figure 10 sensors-24-00645-f010:**
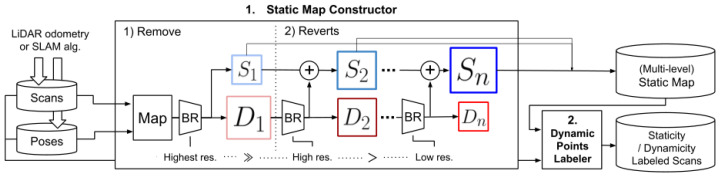
The pipeline of the *Remove, then Revert* method.

**Figure 11 sensors-24-00645-f011:**
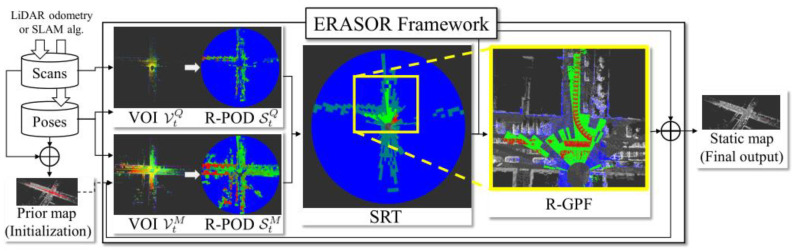
Overview of the *ERASOR* method.

**Table 1 sensors-24-00645-t001:** Abbreviations for terms.

Full Name	Abbreviation
Simultaneous Localization and Mapping	SLAM
Laser Detection and Ranging	LiDAR
Unmanned Aerial Vehicles	UAV
LiDAR Odometry and Mapping	LOAM
LiDAR-Inertial Odometry	LIO
Dempster Shafer Theory	DST
Aggregate View Object Detection	AVOD
Field of View	FOV
Viewpoint Feature Histogram	VFH
Random Sample Consensus	RANSAC
Convolutional Neural Network	CNN
Region Proposal Network	RPN
Recurrent Neural Network	RNN
Speeded Up Robust Features	SURF
Conditional Random Field	CRF
Implicit Moving Least Squares	IMLS
Region-wise Ground Plane Fitting	R-GPF
Principal Component Analysis	PCA
Normal Distribution Transformation	NDT

**Table 2 sensors-24-00645-t002:** Ray-tracing-based approach.

Year	Author	Main Idea	Problem
2012	Azim et al. [51]	Pay attention to the motion characteristics of objects	Poor robustness in chaotic environments
2013	Underwood et al. [52]	Use ray tracing of points in a spherical coordinate system	Can only compare two scans at a time
2013	Hornung et al. [38]	Use octree data structure to store occupancy information	The probability update function is very sensitive
2015	Xiao et al. [54]	Fusion of point-to-face distance information	The processing of the dynamic points cloud is not voxelization
2016	Postica et al. [55]	Propose an image-based verification step	Ignore the point cloud beyond the 30m range of the sensor
2016	Asvadi et al. [57]	Adopt a ray-tracing-based data structure	Need to first extract the plane features of the ground
2016	Chen et al. [59]	Combine models through adaptive weighting	Only applicable to filtering out pedestrians
2017	Gehrung et al. [60]	Use the accumulation of probability mass	Easy to label static background spots as dynamic objects
2018	Schauer et al. [33]	Excellent optimization of false positives and false negatives	Huge consumption of computing resources
2020	Pagad et al. [32]	Combine training neural network with octree grid	The technologies used is relatively complex
2023	Zhang et al. [61]	Noise removal and ground segmentation optimized for the *OctoMap*	High computational resources and slow execution speeds

**Table 3 sensors-24-00645-t003:** Visibility-based approach.

Year	Author	Main Idea	Problem
2014	Pomerleau et al. [40]	Calculate Bayesian probability discriminant dynamic points	Uses a fixed-size association rule
2014	Ambrus et al. [48]	Propose a new system of spatial reasoning	Assumes that the correct position is given
2019	Yoon et al. [35]	Only two scans are needed to determine the dynamic point cloud	Assumes that the pose of the scan is given by SLAM
2020	Kim et al. [34]	False positives revert based on the multi-resolution range images	Limitations when dealing with occlusion
2021	Qian et al. [63]	Filtering of dynamic objects prior to front-end alignment	Slightly less accurate at filtering dynamic point cloud
2021	Chen et al. [64]	Combining visibility with semantic segmentation of point clouds	Misses some dynamic objects that are temporarily stationary

**Table 4 sensors-24-00645-t004:** Segmentation-based approach.

Year	Author	Main Idea	Problem
2009	Petrovskaya et al. [70]	Model vehicles as two-dimensional bounding boxes	Use hand-made models rather than learning
2010	Shackleton et al. [72]	Spatial segmentation of point clouds using 3D grids	Not suitable for use with mobile 3D laser sensors
2011	Sprecher et al. [73]	Background subtraction using accessibility analysis	
2012	Kaestner et al. [71]	Propose a generation-based object detection algorithm	
2012	Litomisky et al. [74]	Distinguish dynamic clusters within static clusters using VFH	Outlier dynamic points are prone to omitting
2015	Yin et al. [75]	Extraction of dynamic clusters using Euclidean clustering	
2019	Yoon et al. [35]	Propose a region-based growth method	
2017	Chen et al. [76]	Detect dynamic objects based on multi-view 3D networks	Cannot detect untrained objects and occasionally fails to detect objects
2018	Zhou et al. [77]	Propose a voxel-coding approach for extracting features	
2018	Ruchti et al. [79]	Neural networks to predict the probability of dynamic objects	
2019	Shi et al. [78]	Segmentation of point clouds via PointNet++ networks	
2018	Wu et al. [82]	Transform point clouds into image form as input to CNN	Relies on manually labeled training data, and the ability to detect new objects are limited.
2019	Zhao et al. [81]	Construct the segmentation network using dense matrix coding	
2019	Milioto et al. [65]	Segmentation using CNN combined with range images	
2019	Biasutti et al. [84]	Propose a U-Net based network for semantic segmentation	
2020	Cortinhal et al. [66]	Propose the SalsaNext method	
2018	Yu et al. [46]	Combine segmentation networks with consistency checking	Relies on manually labeled, high-quality data sets
2018	Sun et al. [88]	Model each cell as an RNN	
2021	Chen et al. [64]	Use continuous range images as an intermediate representation	
2022	He et al. [90]	Use AR-SI theory to improve the accuracy of moving object recognition	
2022	Maneekwan et al. [92]	Combining dynamic object segmentation with local environment prediction	
2023	Mersch et al. [89]	Use 4D CNN to jointly extract spatio-temporal features	

**Table 5 sensors-24-00645-t005:** Other approaches.

Year	Author	Main Idea	Problem
2013	Moosmann et al. [93]	Propose a joint self-positioning and object tracking method	Assumes that the object can be tracked in subsequent scans
2016	Dewan et al. [94]	Use rigid scene flow to detect dynamic objects	Dependent on the minimum speed assumption
2017	Ushani et al. [97]	Learning-based approach to compute scene flows	Assumes the motion of objects is confined to a horizontal plane
2018	Deschaud et al. [96]	Propose a scan-to-model-based matching framework	Assumes that all objects smaller than a size are dynamic objects
2021	Lim et al. [31]	Compare point cloud height to detect dynamic objects	Assumes dynamic objects are all in contact with the ground
2023	Wang et al. [98]	Proposed a vertical voxel height descriptor for online processing	Less localization accuracy and robustness under sloping pavement

**Table 6 sensors-24-00645-t006:** Different methods using the online real-time filtering out of dynamic objects strategy.

Approach	Author	Main Idea	Problem
In the front-end	Qian et al. [63]	Apply a visibility-based approach to the front-end	Limited by the shortcomings of Visibility-based approach
	Pfreundschuh et al. [102]	Generate unsupervised datasets for training based on deep learning	Dynamic objects that belong to untrained classes cannot be recognized
In the back-end	Yoon et al. [35]	Dynamic point cloud filtering only requires two reference scans	Limited reference scans result in slightly poor filtering accuracy
	Fan et al. [103]	Speed up the ray-tracing process and reduce resource consumption	

**Table 7 sensors-24-00645-t007:** Different dynamic point cloud filtering methods using post-processing strategy.

Method	Author	Advantage	Disadvantage
*The peopleremover* (Ray-tracing-based)	Schauer et al. [33]	Excellent optimization of false positives and false negatives	Huge consumption of computing resources
*Remove, then Revert* (Visibility-based)	Kim et al. [34]	Optimization of the most serious false-positive problem	Limitations when dealing with occlusion
*ERASOR* (Other approaches)	Hyungtae et al. [31]	The limitations of the above two methods are avoided	Less robust in complex dynamic environments

**Table 8 sensors-24-00645-t008:** Related works for long-term strategy.

Year	Author	Main Idea	Problem
2014	Pomerleau et al. [40]	Unify dynamic object filtering with map updates	Assumes the poses between query scan and map are exact
2013	Tipaldi et al. [109]	Propose an online update method for 2D grid maps	Other forms of map representation are less suitable for change detection in 3D environments
2015	Einhorn et al. [108]	Combine occupancy grid maps with NDT maps	
2021	Zhao et al. [4]	Introduce the concept of SLAM sessions	More concerned with real-time updating than dynamic objects detecting
2021	Kim et al. [101]	Give attention to both track alignment and map fusion	Lack of guidance for real-time localization

**Table 9 sensors-24-00645-t009:** Different strategies for dynamic object filtering.

Strategy	Applicable Objects	Advantage	Disadvantage
Online real-time filtering	High dynamic objects	Good synchronization	Limited reference scans lead to slightly poor accuracy
Post-processing after mapping	High and low dynamic objects	Achieves more accurate and thorough filtering	Dynamic object filtering lags after the SLAM finished
Long-term SLAM	All dynamic objects	Detects all dynamic objects between different sessions	Difficulty in trajectory alignment and map fusion

## Data Availability

Data are contained within the article.

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
