# Peer review of "A Review of Dynamic Object Filtering in SLAM Based on 3D LiDAR"

_sensors, 2024, doi:10.3390/s24020645_

Round 1
Reviewer 1 Report (Previous Reviewer 1)
Comments and Suggestions for Authors First of all, I would like to apologize for the scope of the special issue, something that logically I should have checked beforehand. Undoubtedly, LiDAR and RADAR have similarities the fundamental difference is the wavelength, but it is also true, that the methodologies of use and processing are markedly different, and even at the level of sensors and platforms used, at least for cartographic purposes. The manuscript is a bibliographic review, so the comments are oriented to its interest, rather than oriented to evaluate other issues.
However, focusing on the comments to the manuscript submitted for evaluation, in my opinion, I consider that although it is not a usual research work, in which new developments are presented, at the level of instruments or methodologies, in the same, if a deep and careful review on the different methodologies related to the filtering of objects in SLAM LiDAR point clouds is performed. In this sense, I believe that the work can have an undoubted usefulness for researchers who are being incorporated into the use of these methodologies. The manuscript is well structured, incorporating details about the problem to be solved and which are the most common methodologies for it. All this, accompanied by a large number of bibliographical references (current, as it could not be otherwise considering the subject matter in question) will allow the reader to have a greater depth in some of the specific topics. I particularly consider it a useful contribution for publication in the journal, for which I thank the authors for their work.
Author Response
Please see the attachment.

Reviewer 2 Report (Previous Reviewer 3)
Comments and Suggestions for Authors
In the first revision, I asked you to remove the term “etc”, so you replaced it with “and so on”. This is not correct. You should use “such as”, e.g., High dynamic objects: objects that move continuously in the LiDAR scan, such as 588 people walking on the street, moving vehicles, running pets and so on. è High dynamic objects: objects that move continuously in the LiDAR scan, such as people walking on the street, moving vehicles, and running pets.
Please check completely the paper.
Comments on the Quality of English Language
Minor editing of English language required.
Author Response
Please see the attachment

This manuscript is a resubmission of an earlier submission. The following is a list of the peer review reports and author responses from that submission.
Round 1
Reviewer 1 Report
Comments and Suggestions for Authors
First of all, I would like to thank the Editorial Board of Sensor for the opportunity to participate in the review of this manuscript.
Actually, I should first point out an element to be considered by the Editorial Board of the journal, in the sense that the manuscript addresses a literature review on LiDAR-centered methodologies when the special issue addresses RADAR topics. In this sense, my recommendation should be that the manuscript be redirected to another special issue with a more appropriate theme, so that the work achieves the dissemination it deserves.
As for the work, as I mentioned it is not really a research work oriented to own developments, but it is oriented to a methodological review linked to the methods of filtering moving targets in scenes captured with 3D LiDAR systems with the aim of improving the application of SLAM techniques. In this sense, I consider that the subject matter is current and interesting and addresses a specific and important problem, and will surely be of interest to many readers of Sensors. Logically, the evaluation scheme of a paper of this nature should be different in the sense that it does not have its own contributions.
In my opinion the work is well structured, and although it is a little long (27 pages) it is logical considering the topic addressed, also and as an aspect of value, indicate that it provides a total of 101 bibliographical references that provide additional information on the different methods analyzed, results and applications of the same. The text is well written and the presentation is easy to follow.
In this sense, my recommendation would be to reject the work for publication in this special issue, not because of a lack of quality, but because it does not fit the specific subject matter of this issue. And this recommendation is accompanied by an acceptance of the manuscript in another issue that addresses problems related to the capture of information using 3D LiDAR methodologies and specifically related to the classification of terrestrial scenes and / or application of SLAM.
Reviewer 2 Report
Comments and Suggestions for Authors
Very well organized and rather thorough review of dynamic object removal. Essentials of each method are provided, also the historical development has been covered in an interesting way, which fits this topic well.
Details:
Several oversized tables could be nurtured to fit the margins better (e.g. arraging more lines per box).
p.21 As can be seen , the focus of ??? is not omn detectting ... --> fix the citation.
Reviewer 3 Report
Comments and Suggestions for Authors
In the reference list there are no references 2023, normally, a review study must consider the more recent studies. Please update the reference list and then the paper.
Abbreviations must be defined in the abstract and in the rest of paper. How we define an abbreviation, e.g., Simultaneous Localization and Mapping (SLAM). Please check that all used abbreviations in abstract are defined within the abstract, and all used abbreviations in the rest of paper are defined also in the first passage in the paper.
Abstract
Line 11: “3D point cloud map” must be “3D point cloud”.
Please avoid using (we, our and us), use the passive voice. Please check the paper.
The abstract does not represent an abstract, please rewrite it focusing on the suggested approach to classify the Dynamic Object Filtering in SLAM methods. The importance of this approach.
How we write “lidar”: LiDAR, lidar, or Lidar, no one write it “LIDAR”
Light Detection And Ranging (LiDAR).
Introduction
Line 37: “the time-of-flight of the reflected beams to create a 3D point cloud of the environment”è Some systems measure the phase difference in addition to the travel time, please revise this idea from the literature and correct the sentence.
Lines 38, 39, and 40: The point cloud represents the geometry and spatial layout of the environment, which can be used to build a map 39 and localize the robot within it. How can the point cloud represent the geometry? Please cite reference for that, or prove it, or correct it.
Line 40: “Over the years” which years? Please you write a scientific paper that is why you must be specific. Then you add references 5 (2012) and 6 (2019), and the two references are old relatively,
Here you talk about general LiDAR, so you can add modern papers such as:
Dey, E., Awrangjeb, M., Tarsha Kurdi, F., Stantic, B., 2023. Machine learning-based segmentation of aerial LiDAR point cloud data on building roof. European Journal of Remote Sensing, 56: 1, doi: 10.1080/22797254.2023.2210745.
Wang, Y., Lin, Y., Cai, H., Li, S. 2023. Hierarchical Fine Extraction Method of Street Tree Information from Mobile lidar Point Cloud Data. Applied Sciences 2023, Vol. 13, Page 276, 13(1), 276. https://doi.org/10.3390/APP13010276.
Please remove Figure 2.
Remove Table 1 from the paper and add it the appendix section.
As there is an overlap between static and dynamic point cloud filtering, it will be benefit to the readers to refer to recent review papers of static and/ or dynamic point cloud processing such as:
Gharineiat, Z., Tarsha Kurdi, F., Campbell, G. 2022. Review of automatic processing of topography and surface feature identification LiDAR data using machine learning techniques. Remote Sens. 2022, 14 (19), 4685, https://doi.org/10.3390/rs14194685.
Mirzaei, K.; Arashpour, M.; Asadi, E.; Masoumi, H.; Bai, Y.; Behnood, A. 3D point cloud data processing with machine learning for construction and infrastructure applications: A comprehensive review. Advanced Engineering Informatics, vol 51, 101501, ISSN 1474-0346, 2022, https://doi.org/10.1016/j.aei.2021.101501.
Dynamic target point filtering methods in 3D point cloud mapsè must be “Dynamic target point cloud filtering”.
Attention, the point cloud does not represent a map, please check all the text.
Voxel-based approach
This type of method is based on laser ray tracing [45]è this phrase is not correct, because [45] use the Voxel for ray tracing and not the opposite.
“voxel grids [46] (or octree girds” is the voxel grid is the same of octree grid? I don’t believe that.
Figure 4 is not correct, there are no voxels, and simple idea that you want to explain becomes vague with this figure. Please remove it.
Line 130: please erase the phrase “Azim and Aycard were among the pioneers to employ the voxel-based approach for dynamic object filtering in SLAM based on 3D LiDAR” , please don’t use the word “pioneer”, check all the paper.
In Tables 2, 3, 4, 5, and 6, please replace the word “Innovation” by “Main idea”. In Author column, add the cited reference number.
Please remove Figure 6 because it is useless.
Reference citation in the text does not respect the journal author guideline, please check all reference citation after revision the journal guideline.
Line 197: “Schauer et al. creatively proposed the method”: all suggested approach in the literature represent a creative work, hence you should erase the word “creatively”. Please check all the paper for similar cases.
Line 280: you use the term “etc.”, please remove it, and check all the paper, you have not to use it.
Line 411: “novel approach” Please remove the word “novel” and check the paper.
Processing strategies of dynamic objects in the SLAM process: it is unacceptable to use the world “Processing” because it represents a general term and you must be precise.
Line 470: “we propose three distinct processing strategies:” you don’t suggest them because make a review, you can say “three ???????? strategies are proposed in the literature”, do you mean that?
Please rewrite Section 3 to replace the word processing by clear and accurate expression. Listen, in laser scanning there are several envisaged tasks such as filtering, classification, segmentation, feature extraction, modelling, downsampling, etc, all these operations are processing. One of these operations consists of a list of consecutive procedures, but every procedure has also an accurate name. you must be precise.
Section 3.1 is out of paper topic, if you want to keep it, you should put it in the introduction section.
Section 3.2 seems that you start navigating out of the review paper, there are no references, nor suggested approaches. Do you believe that this idea needs an independent section?
Section 3.3 was presented before in Section 2, why you didn’t integrate it in Section2?
The paper structure is not clear for me, please provide in Section 1 a workflow chart that describe the target topic elements as they will be handled in the paper. Then the paper structure will be built according to this organigram.
Comments on the Quality of English Language
Minor editing of English language required